# IEPT: Instance-Level and Episode-Level Pretext Tasks for Few-Shot Learning

**Manli Zhang, Jianhong Zhang & Zhiwu Lu** *
Gaoling School of Artificial Intelligence, Renmin University of China, Beijing, China
Beijing Key Laboratory of Big Data Management and Analysis Methods, Beijing, China
{manlizhang,jianhong,luzhiwu}@ruc.edu.cn

**Tao Xiang**
University of Surrey, Guildford, Surrey, UK
t.xiang@surrey.ac.uk

**Mingyu Ding**
The University of Hong Kong, Hong Kong
mingyuding@hku.hk

**Songfang Huang**
Alibaba DAMO Academy, Hangzhou, China
songfang.hsf@alibaba-inc.com

## Abstract

The need of collecting large quantities of labeled training data for each new task has limited the usefulness of deep neural networks. Given data from a set of source tasks, this limitation can be overcome using two transfer learning approaches: few-shot learning (FSL) and self-supervised learning (SSL). The former aims to learn 'how to learn' by designing learning episodes using source tasks to simulate the challenge of solving the target new task with few labeled samples. In contrast, the latter exploits an annotation-free pretext task across all source tasks in order to learn generalizable feature representations. In this work, we propose a novel Instance-level and Episode-level Pretext Task (IEPT) framework that seamlessly integrates SSL into FSL. Specifically, given an FSL episode, we first apply geometric transformations to each instance to generate extended episodes. At the instance-level, transformation recognition is performed as per standard SSL. Importantly, at the episode-level, two SSL-FSL hybrid learning objectives are devised: (1) The consistency across the predictions of an FSL classifier from different extended episodes is maximized as an episode-level pretext task. (2) The features extracted from each instance across different episodes are integrated to construct a single FSL classifier for meta-learning. Extensive experiments show that our proposed model (i.e., FSL with IEPT) achieves the new state-of-the-art.

## 1 Introduction

Deep convolutional neural networks (CNNs) (Krizhevsky et al., 2012; He et al., 2016b; Huang et al., 2017) have seen tremendous successes in a wide range of application fields, especially in visual recognition. However, the powerful learning ability of CNNs depends on a large amount of manually labeled training data. In practice, for many visual recognition tasks, sufficient manual annotation is either too costly to collect or not feasible (e.g., for rare object classes). This has severely limited the usefulness of CNNs for real-world application scenarios. Attempts have been made recently to mitigate such a limitation from two distinct perspectives, resulting in two popular research lines, both of which aim to transfer knowledge learned from the data of a set of source tasks to a new target one: few-shot learning (FSL) and self-supervised learning (SSL).

FSL (Fei-Fei et al., 2006; Vinyals et al., 2016; Finn et al., 2017; Snell et al., 2017; Sung et al., 2018) typically takes a 'learning to learn' or meta-learning paradigm. That is, it aims to learn an algorithm for learning from few labeled samples, which generalizes well across any tasks. To that end, it adopts

---

*Corresponding author.

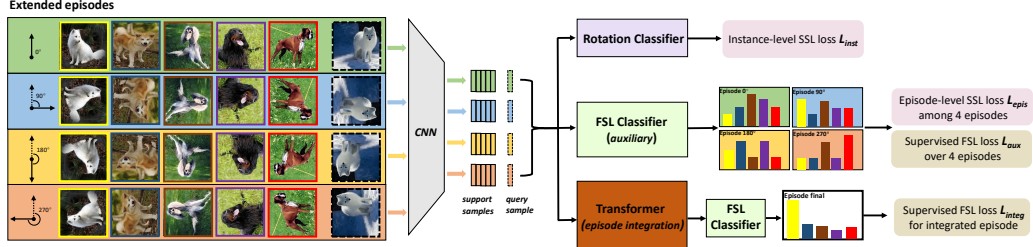

Figure 1: Schematic of our approach to FSL. Given a training episode, we apply 2D rotations by 0, 90, 180, and 270 degrees to each instance to generate four extended episodes. After going through a feature extraction CNN, four losses over three branches are designed: (1) In the top branch, we employ a self-supervised rotation classifier with the instance-level SSL loss $\mathcal{L}_{inst}$. (2) In the middle branch, an FSL classifier is exploited to predict the FSL classification probabilities for each episode. We maximize the classification consistency among the extended episodes by forcing the four probability distributions to be consistent using $\mathcal{L}_{epis}$. The average supervised FSL loss $\mathcal{L}_{aux}$ is also computed. (3) In the bottom branch, we utilize an integration transformer module to fuse the features extracted from each instance with different rotation transformations; they are then used to compute an integrated FSL loss $\mathcal{L}_{integ}$. Among the four losses, $\mathcal{L}_{inst}$ and $\mathcal{L}_{epis}$ are the self-supervised losses, and $\mathcal{L}_{aux}$ and $\mathcal{L}_{integ}$ are the supervised losses.

an episodic training strategy – the source tasks are arranged into learning episodes, each of which contains $n$ classes and $k$ labeled samples per class to simulate the setting for the target task. Part of the CNN model (e.g., feature extraction subnet, classification layers, or parameter initialization) is then meta-learned for rapid adaptation to new tasks.

In contrast, SSL (Doersch et al., 2015; Noroozi & Favaro, 2016; Iizuka et al., 2016; Doersch & Zisserman, 2017; Noroozi et al., 2018) does not require the source data to be annotated. Instead, it exploits an annotation-free pretext task on the source task data in the hope that a task-generalizable feature representation can be learned from the source tasks for easy adoption or adaptation in a target task. Such a pretext task gets its self-supervised signal at the per-instance level. Examples include rotation and context prediction (Gidaris et al., 2018; Doersch et al., 2015), jigsaw solving (Noroozi & Favaro, 2016), and colorization (Iizuka et al., 2016; Larsson et al., 2016). Since these pretext tasks are class-agnostic, solving them leads to the learning of transferable knowledge.

Since both FSL and SSL aim to reduce the need of collecting a large amount of labeled training data for a target task by transferring knowledge from a set of source tasks, it is natural to consider combining them in a single framework. Indeed, two recent works (Gidaris et al., 2019; Su et al., 2020) proposed to integrate SSL into FSL by adding an auxiliary SSL pretext task in an FSL model. It showed that the SSL learning objective is complementary to that of FSL and combining them leads to improved FSL performance. However, in (Gidaris et al., 2019; Su et al., 2020), SSL is combined with FSL in a superficial way: it is only taken as a separate auxiliary task for each single training instance and has no effect on the episodic training pipeline of the FSL model. Importantly, by ignoring the class labels of samples, the instance-level SSL learning objective is weak on its own. Since meta-learning across episodes is the essence of most contemporary FSL models, we argue that adding instance-level SSL pretext tasks alone fails to exploit fully the complementarity of the aforementioned FSL and SSL, for which a closer and deeper integration is needed.

To that end, in this paper we propose a novel Instance-level and Episode-level Pretext Task (IEPT) framework for few-shot recognition. Apart from adding an instance-level pretext SSL task as in (Gidaris et al., 2019; Su et al., 2020), we introduce two episode-level SSL-FSL hybrid learning objectives for seamless SSL-FSL integration. Concretely, as illustrated in Figure 1, our full model has three additional learning objectives (besides the standard FSL one): (1) Different rotation transformations are applied to each original few-shot episode to generate a set of extended episodes, where each image has a rotation label for the instance-level pretext task (i.e., to predict the rotation label). (2) The consistency across the predictions of an FSL classifier from different extended episodes is maximized as an episode-level pretext task. For each training image, the rotation transformation does not change its semantic content and hence its class label; the FSL classifier predictions across different extended episodes thus should be consistent, hence the consistency regularization objective. (3) The correlation of features across instances from these extended episodes is modeled by a

transformer-based attention module, optimizing the fusion of the features of each instance/image and its various rotation-transformed versions mainly for task adaptation during meta-testing. Importantly, with these three new learning objectives introduced in IEPT, any meta-learning based FSL model can now benefit more from SSL by fully exploiting their complementarity.

Our main contributions are: (1) For the first time, we propose both instance-level and episode-level pretext tasks (IEPT) for integrating SSL into FSL. The episode-level pretext task enables episodic training of SSL and hence closer integration of SSL with FSL. (2) In addition to these pretext tasks, FSL further benefits from SSL by integrating features extracted from various rotation-transformed versions of the original training instances. The optimal way of feature integration is learned by a transformer-based attention module, which is mainly designed for task adaptation during meta-testing. (3) Extensive experiments show that the proposed model achieves the new state-of-the-art.

## 2 Related Work

**Few-Shot Learning.** The recent FSL studies are dominated by meta-learning based methods. They can be divided into three groups: (1) Metric-based methods (Vinyals et al., 2016; Snell et al., 2017; Sung et al., 2018; Allen et al., 2019; Xing et al., 2019; Li et al., 2019a;b; Wu et al., 2019; Ye et al., 2020; Afrasiyabi et al., 2020; Liu et al., 2020; Zhang et al., 2020) aim to learn the distance metric between feature embeddings. The focus of these methods is often on meta-learning of a feature-extraction CNN, whilst the classifiers used are of simple form such as a nearest-neighbor classifier. (2) Optimization-based methods (Finn et al., 2017; Ravi & Larochelle, 2017; Rusu et al., 2019; Lee et al., 2019) learn to optimize the model rapidly given a few labeled samples per class in the new task. (3) Model-based methods (Santoro et al., 2016; Munkhdalai & Yu, 2017; Mishra et al., 2018) focus on designing either specific model structures or parameters capable of rapid updating. Apart from these three groups of methods, other FSL methods have attempted feature hallucination (Schwartz et al., 2018; Hariharan & Girshick, 2017; Gao et al., 2018; Wang et al., 2018; Zhang et al., 2019; Tsutsui et al., 2019) which generates additional samples from the given few shots for network finetuning, and parameter predicting (Qiao et al., 2018; Qi et al., 2018; Gidaris & Komodakis, 2019; 2018) which learns to predict part of the parameters of a network given few samples of new classes for quick adaptation. In this work, we adopt the metric-based Prototypical Network (ProtoNet) (Snell et al., 2017) as the basic FSL classifier for the main instantiation of our IEPT framework due to its simplicity and popularity. However, we show that any meta-learning based FSL method can be combined with our IEPT (see results in Figure 2(c)).

**Self-Supervised Learning.** In SSL, it is assumed that the source task data is label-free and a pretext task is designed to provide self-supervision signals at the instance-level. Existing SSL approaches differ mainly in the pretext task design. These include predicting the rotation angle (Gidaris et al., 2018) and the context of image patch (Doersch et al., 2015; Nathan Mundhenk et al., 2018), jigsaw solving (Noroozi & Favaro, 2016; Noroozi et al., 2018) (i.e. shuffling and then reordering image patch), and performing images reversion (Iizuka et al., 2016; Pathak et al., 2016; Larsson et al., 2016). SSL has been shown to be beneficial to various down-steam tasks such as semantic object matching (Novotny et al., 2018), object segmentation (Ji et al., 2019) and object detection (Doersch & Zisserman, 2017) by learning transferable feature presentations for these tasks.

**Integrating Self-Supervised Learning into Few-Shot Learning.** To the best of our knowledge, only two recent works (Gidaris et al., 2019; Su et al., 2020) have attempted combining SSL with FSL. However, the integration of SSL into FSL is often shallow: the original FSL training pipeline is intact; in the meantime, an additional loss on each image w.r.t. a self-supervised signal like the rotation angle or relative patch location is introduced. With pretext tasks solely at the instance level, combining the two approaches (i.e., SSL and FSL) can only be superficial without fully exploiting the episodic training pipeline unique to FSL. Different from (Gidaris et al., 2019; Su et al., 2020), we introduce an episode-level pretext task to integrate SSL into the episodic training in FSL fully. Specifically, the consistency across the predictions of an FSL classifier from different extended episodes is maximized to reflect the fact that various rotation transformations should not alter the class-label prediction. Moreover, features of each instance and its various rotation-transformed versions are now fused for FSL classification, to integrate SSL with FSL for the supervised classification task. Our experimental results show that thanks to the closer integration of SSL and FSL, our IEPT clearly outperforms (Gidaris et al., 2019; Su et al., 2020) (see Table 1).

## 3 METHODOLOGY

### 3.1 PRELIMINARY

**Problem Setting.** Given an $n$-way $k$-shot FSL task sampled from a test set $\mathcal{D}_t$, to imitate the test setting, an FSL model is typically trained in an episodic way. That is, $n$-way $k$-shot episodes are randomly sampled from a training set $\mathcal{D}_s$, where the class label space of $\mathcal{D}_s$ has no overlap with that of $\mathcal{D}_t$. Each episode $E_e$ contains a support set $\mathcal{S}_e$ and a query set $\mathcal{Q}_e$. Concretely, we first randomly sample a set of $n$ classes $\mathcal{C}_e$ from the training set, and then generate $\mathcal{S}_e$ and $\mathcal{Q}_e$ by sampling $k$ support samples and $q$ query samples from each class in $\mathcal{C}_e$, respectively. Formally, we have $\mathcal{S}_e = \{(x_i, y_i)|y_i \in \mathcal{C}_e, i = 1, ..., n \times k\}$ and $\mathcal{Q}_e = \{(x_i, y_i)|y_i \in \mathcal{C}_e, i = 1, ..., n \times q\}$, where $\mathcal{S}_e \bigcap \mathcal{Q}_e = \emptyset$. For simplicity, we denote $l_k = n \times k$ and $l_q = n \times q$. In the meta-training stage, the training process has an inner and an outer loop in each episode: in the inner loop, the model is updated using $\mathcal{S}_e$; its performance is then evaluated on the query set $\mathcal{Q}_e$ in the outer loop to update the model parameters or algorithm that one wants to meta-learn.

**Basic FSL Classifier.** We employ ProtoNet (Snell et al., 2017) as the basic FSL model. This model has a feature-extraction CNN and a simple non-parametric classifier. The parameter of the feature extractor is to be meta-learned. Concretely, in the inner loop of an episode, ProtoNet fixes the feature extractor and computes the mean feature embedding for each class as follows:

$$h_c = \frac{1}{k} \cdot \sum_{(x_i, y_i) \in \mathcal{S}_e} f_\phi(x_i) \cdot \mathbb{I}(y_i = c), \tag{1}$$

where class $c \in \mathcal{C}_e$, $f_\phi$ is a feature extractor with learnable parameters $\phi$, and $\mathbb{I}$ is the indicator function. By computing the distance between the feature embedding of each query sample and that of the corresponding class, the loss function used to meta-learn $\phi$ in the outer loop is defined as:

$$\mathcal{L}_{fsl}(\mathcal{S}_e, \mathcal{Q}_e) = \frac{1}{|\mathcal{Q}_e|} \sum_{(x_i, y_i) \in \mathcal{Q}_e} - \log \frac{\exp(-d(f_\phi(x_i), h_{y_i}))}{\sum_{c \in \mathcal{C}_e} \exp(-d(f_\phi(x_i), h_c))}, \tag{2}$$

where $d(\cdot, \cdot)$ denotes a distance function (e.g., the $l_2$ distance).

### 3.2 PRETEXT TASKS IN IEPT

The schematic of our IEPT is illustrated in Figure 1. We first define a set of 2D-rotation operators $\mathcal{G} = \{g_r | r = 0, ..., R - 1\}$, where $g_r$ means the operator of rotating the image by $r*90$ degrees and $R$ is the total number of rotations ($R = 4$ in our implementation). Given an original episode $E_e = \{\mathcal{S}_e, \mathcal{Q}_e\}$ as described in Sec. 3.1, we utilize the 2D-rotation operators from $\mathcal{G}$ in turn to transform each image in $E_e$. This results in a set of $R$ extended episodes (including the original one) $E = \{\{\mathcal{S}_e^r, \mathcal{Q}_e^r\} | r = 0, ..., R - 1\}$, where $\mathcal{S}_e^r = \{(x_i, y_i, r)|y_i \in C_e, i = 1, ..., l_k\}$ and $\mathcal{Q}_e^r = \{(x_i, y_i, r)|y_i \in C_e, i = 1, ..., l_q\}$. Now each episode is denoted as $E_e^r = \{(x_i, y_i, r)|y_i \in C_e, i = 1, ..., l_k, l_k + 1, ..., l_k + l_q\}$, where the first $l_k$ samples are from $\mathcal{S}_e^r$ and the rest from $\mathcal{Q}_e^r$. Note that $\{\mathcal{S}_e^0, \mathcal{Q}_e^0\}$ is the original episode $\{\mathcal{S}_e, \mathcal{Q}_e\}$. With the rotation transformations, each sample $(x_i, y_i, r_i)$ in $E$ carries a class label $y_i$ for supervised learning (from the inherent class) and a label $r_i$ from the rotation operator for self-supervised learning. After generating the set of extended episodes $E$, the feature extractor $f_\phi$ is applied to each image $x_i$ in $E$. On these episodes, we design two self-supervised pretext tasks, one at the instance-level and the other episode-level.

**Instance-Level Pretext Task.** The instance-level task is to recognize different rotation transformations. The idea is that if the model to be meta-learned here (i.e., $f_\phi$) can be used to distinguish different transformations, it must understand the canonical poses of objects (e.g., animals have legs touching the ground and trees have leaves on top), a vital part of class-agnostic and thus transferable knowledge. With the self-supervised rotation label $r_i$, we consider the mapping: $f_{\theta_{rot}} : x_i \mapsto r_i$ for each instance $(x_i, y_i, r_i) \in E$, where $f_{\theta_{rot}}$ is a rotation classifier with learnable parameters $\theta_{rot}$. Given the input pair $(x_i, r_i)$, the total instance-level rotation loss is a cross-entropy loss:

$$\mathcal{L}_{inst} = \frac{1}{R(l_k + l_q)} \sum_{r=0}^{R-1} \sum_{(x_i, y_i, r_i) \in E_e^r} - \log \frac{\exp([f_{\theta_{rot}}(f_\phi(x_i))]_{r_i})}{\sum_{r'=0}^{R-1} \exp([f_{\theta_{rot}}(f_\phi(x_i))]_{r'})}, \tag{3}$$

where $[f_{\theta_{rot}}(f_\phi(x_i))] \in \mathbb{R}^R$ is the rotation scoring vector and $[\cdot]_r$ means taking the $r$-th element.

**Episode-Level Pretext Task.** We design the episode-level task based on a simple principle: although different extended episodes contain images with different rotation transformations, these transformations do not change their class labels. Consequently, the FSL classifier should produce consistent probability distributions for each instance across different extended episodes. Such consistency can be measured using the Kullback–Leibler (KL) divergence. Formally, for each extended episode $\{\mathcal{S}_e^r, \mathcal{Q}_e^r\}$ in $E$, we first define the probability distribution of FSL classification over the query set $\mathcal{Q}_e^r$ as $\mathcal{P}_e^r = [p_1^r; \cdots; p_{l_q}^r] \in \mathbb{R}^{l_q \times n}$, where $p_i^r \in \mathbb{R}^n$ is the probability distribution for $x_i$ in $\mathcal{Q}_e^r$ with its $c$-th element $[p_i^r]_c$ $(c = 1, ..., n)$ being:

$$[p_i^r]_c = \frac{\exp(-d(f_\phi(x_i), h_c^r))}{\sum_{c'} \exp(-d(f_\phi(x_i), h_{c'}^r))}. \tag{4}$$

The above probability is computed as in Sec. 3.1 and the class embedding $h_c^r$ is obtained from $\mathcal{S}_e^r$. The mean probability distribution of the $R$ extended episodes is thus given by:

$$\hat{p}_i = \frac{1}{R} \cdot \sum_{r=0}^{R-1} p_i^r. \tag{5}$$

The total episode-level consistency regularization loss is computed with the KL divergence loss:

$$\mathcal{L}_{epis} = \frac{1}{Rl_q} \cdot \sum_{r=0}^{R-1} \sum_{i=1}^{l_q} \text{mean}(p_i^r(\log p_i^r - \log \hat{p}_i)). \tag{6}$$

where $\text{mean}(\cdot)$ is an element-wise averaging function.

## 3.3 INTEGRATED FSL TASK

The two tasks introduced so far are self-supervised tasks without using the class labels in the query set. Now we describe how in the supervised classification task, the extended episodes can be used.

Given the set of extended episodes $E$, we denote the feature set of $E$ as $E_{emb}$, where $E_{emb} = \{f_\phi(x_i)|(x_i, y_i, r) \in E_e^r, r = 0, \cdots, R-1, i = 1, ..., l_k + l_q\}$. Note that each extended episode in $E$ corresponds to one specific rotation transformation of the same set of images from the original episode $E_e$. Therefore, in order to capture the correlation among instances with different transformations and learn how best combine them to form the class mean for meta-learning, an instance attention module is deployed w.r.t. each image in $E_e$ (i.e., all images are assumed to be independent). Specifically, based on $E_{emb}$, we construct the feature tensor $F \in \mathbb{R}^{(l_k+l_q) \times R \times d}$, where $d$ is the feature dimension. We then adopt a transformer to obtain the integrated representation for FSL classification. The transformer architecture is based on a self-attention mechanism, as in (Vaswani et al., 2017). It receives the triplet input $(F, F, F)$ as $(Q, K, V)$ (Query, Key, and Value, respectively). With $F^{(i)}$ being the $i$-th row of $F$ (w.r.t. the $i$-th image in $E_e$), the attentive module is defined as:

$$(F_Q^{(i)}, F_K^{(i)}, F_V^{(i)}) = (F^{(i)}W_Q, \ F^{(i)}W_K, \ F^{(i)}W_V), \tag{7}$$

$$F_{att}^{(i)} = F^{(i)} + \text{softmax}\Big(\frac{F_Q^{(i)}(F_K^{(i)})^T}{\sqrt{d_K}}\Big) F_V^{(i)}, \tag{8}$$

where $d_K = d$, and $W_Q, W_K, W_V$ represent the parameters of three fully-connected layers respectively (the parameters of the integration transformer are collected as $\theta_{int}$). Note that the key and value are computed from each image and its augmented versions, i.e., they are computed independently without using inter-image correlation. With the attentive feature $F_{att} \in \mathbb{R}^{(l_k+l_q) \times R \times d}$, the integrated representation $F_{integ} = [F^S; F^Q] \in \mathbb{R}^{(l_k+l_q) \times Rd}$ ($F^S$ and $F^Q$ are respectively for the support set and query set) is given by:

$$F_{integ} = \text{flatten}(F_{att}), \tag{9}$$

where $\text{flatten}(\cdot)$ denotes flattening $F_{att}$ along the last two dimensions, i.e., concatenating the attentive features from different extended episodes for the corresponding images. The integrated representation is then inputted to the FSL classifier to define the FSL classification loss:

$$\mathcal{L}_{integ} = \frac{1}{l_q} \cdot \sum_{i=1}^{l_q} -\log \frac{\exp(-d(F_i^Q, h_{y_i}^f))}{\sum_{c \in \mathcal{C}_e} \exp(-d(F_i^Q, h_c^f))} \tag{10}$$

where the class embedding $h_c^f = \frac{1}{k} \cdot \sum_{i=1}^{l_k} F_i^S \cdot \mathbb{I}(y_i = c)$ is computed on the support set. Note that the integrated FSL task actually acts as an alternative to prediction averaging.

### 3.4 TOTAL LOSS

The total training loss for our full model consists of the self-supervised losses from the pretext tasks and the supervised losses from the FSL tasks. In this work, in addition to $\mathcal{L}_{integ}$ in Eq. (10), another supervised FSL loss $\mathcal{L}_{aux}$ is also used (see Figure 1). $\mathcal{L}_{aux}$ is the average FSL classification loss over the extended episodes. Formally, it can be written as:

$$\mathcal{L}_{aux} = \frac{1}{R} \cdot \sum_{r=0}^{R-1} \mathcal{L}_{fsl}(\mathcal{S}_e^r, \mathcal{Q}_e^r) \tag{11}$$

Therefore, the total loss $\mathcal{L}_{total}$ for training our full model is given as follows:

$$\mathcal{L}_{total} = \underbrace{\overbrace{w_1 * \mathcal{L}_{inst}}^{instance-level} + \overbrace{w_2 * \mathcal{L}_{epis}}^{episode-level}}_{\text{self-supervised loss}} + \underbrace{w_3 * \mathcal{L}_{aux} + \mathcal{L}_{integ}}_{\text{supervised loss}}, \tag{12}$$

where $w_1, w_2, w_3$ are the loss weight hyperparameters.

### 3.5 INFERENCE

During the test stage, we only exploit the integrated representation $F_{integ}$ for the final FSL prediction. The predicted class label for $x_i \in \mathcal{Q}_e$ can be computed with Eq. (10) as:

$$y_i^{pred} = \operatorname*{argmax}_{y \in \mathcal{C}_e} \frac{\exp(-d(F_i^Q, h_y^f))}{\sum_{c \in \mathcal{C}_e} \exp(-d(F_i^Q, h_c^f))}. \tag{13}$$

### 3.6 FULL IEPT ALGORITHM

For easy reproduction, we present the full algorithm for FSL with IEPT in Algorithm 1. Once learned, with the learned $\psi$, we can perform the inference over the test episodes with Eq. (13).

---

**Algorithm 1** FSL with IEPT

---

**Input:** The training set $\mathcal{D}_s$, the rotation operator set $\mathcal{G}$
      The loss weight hyperparameters $w_1, w_2, w_3$
**Output:** The learned $\psi$
 1: Randomly initialize all learnable parameters $\psi = \{\phi, \theta_{rot}, \theta_{int}\}$
 2: **for** iteration = 1, ..., MaxIteration **do**
 3:     Randomly sample episode $E_e$ from $\mathcal{D}_s$
 4:     Generate the set of extended episodes $E$ from $E_e$ using $\mathcal{G}$
 5:     Compute the SSL loss $\mathcal{L}_{inst}$ for the instance-level pretext task with Eq. (3)
 6:     Compute the SSL loss $\mathcal{L}_{epis}$ for the episode-level pretext task with Eq. (6)
 7:     Compute the supervised FSL loss $\mathcal{L}_{aux}$ over the extended episodes with Eq. (11)
 8:     Compute the supervised FSL loss $\mathcal{L}_{integ}$ for the integrated episode with Eq. (10)
 9:     $\mathcal{L}_{total} = w_1 * \mathcal{L}_{inst} + w_2 * \mathcal{L}_{epis} + w_3 * \mathcal{L}_{aux} + \mathcal{L}_{integ}$
10:     Update $\psi$ based on $\nabla_\psi \mathcal{L}_{total}$
11: **end for**
12: **return** $\psi$.

---

## 4 EXPERIMENTS

### 4.1 EXPERIMENTAL SETUP

**Datasets.** Two widely-used FSL datasets are selected: *mini*ImageNet (Vinyals et al., 2016) and *tiered*ImageNet (Ren et al., 2018). The first dataset consists of a total number of 100 classes (600 images per class) and the train/validation/test split is set to 64/16/20 classes as in (Ravi & Larochelle, 2017). The second dataset is a larger dataset including 608 classes totally (nearly 1,200 images per class), which is split into 351/97/160 classes for train/validation/test. Both datasets are subsets sampled from ImageNet (Russakovsky et al., 2015).

Table 1: Comparative results for 5-way 1/5-shot FSL. The mean classification accuracies (top-1, %) with the 95% confidence intervals are reported. † indicates the result is reproduced by ourselves.

| Method | Backbone | *mini*ImageNet | | *tiered*ImageNet | |
|---|---|---|---|---|---|
| | | 1-shot | 5-shot | 1-shot | 5-shot |
| MatchingNet (Vinyals et al., 2016) | Conv4-64 | $43.56 \pm 0.84$ | $55.31 \pm 0.73$ | – | – |
| ProtoNet† (Snell et al., 2017) | Conv4-64 | $52.61 \pm 0.52$ | $71.33 \pm 0.41$ | $53.33 \pm 0.50$ | $72.10 \pm 0.41$ |
| MAML (Finn et al., 2017) | Conv4-64 | $48.70 \pm 1.84$ | $63.10 \pm 0.92$ | $51.67 \pm 1.81$ | $70.30 \pm 0.08$ |
| Relation Net (Sung et al., 2018) | Conv4-64 | $50.40 \pm 0.80$ | $65.30 \pm 0.70$ | $54.48 \pm 0.93$ | $71.32 \pm 0.78$ |
| IMP† (Allen et al., 2019) | Conv4-64 | $52.91 \pm 0.49$ | $71.57 \pm 0.42$ | $53.63 \pm 0.51$ | $71.89 \pm 0.44$ |
| DN4 (Li et al., 2019b) | Conv4-64 | $51.24 \pm 0.74$ | $71.02 \pm 0.64$ | – | – |
| DN PARN (Wu et al., 2019) | Conv4-64 | $55.22 \pm 0.84$ | $71.55 \pm 0.66$ | – | – |
| PN+rot (Gidaris et al., 2019) | Conv4-64 | $53.63 \pm 0.43$ | $71.70 \pm 0.36$ | – | – |
| CC+rot (Gidaris et al., 2019) | Conv4-64 | $54.83 \pm 0.43$ | $71.86 \pm 0.33$ | – | – |
| DSN-MR (Simon et al., 2020) | Conv4-64 | $55.88 \pm 0.90$ | $70.50 \pm 0.68$ | – | – |
| Centroid (Afrasiyabi et al., 2020) | Conv4-64 | $53.14 \pm 1.06$ | $71.45 \pm 0.72$ | – | – |
| Neg-Cosine (Liu et al., 2020) | Conv4-64 | $52.84 \pm 0.76$ | $70.41 \pm 0.66$ | – | – |
| IEPT (ours) | Conv4-64 | $\mathbf{56.26 \pm 0.45}$ | $\mathbf{73.91 \pm 0.34}$ | $\mathbf{58.25 \pm 0.48}$ | $\mathbf{75.63 \pm 0.46}$ |
| ProtoNet† (Snell et al., 2017) | Conv4-512 | $53.25 \pm 0.44$ | $73.15 \pm 0.35$ | $57.88 \pm 0.50$ | $76.82 \pm 0.40$ |
| MAML (Finn et al., 2017) | Conv4-512 | $49.33 \pm 0.60$ | $65.17 \pm 0.49$ | $52.84 \pm 0.56$ | $70.91 \pm 0.46$ |
| Relation Net (Sung et al., 2018) | Conv4-512 | $50.86 \pm 0.57$ | $67.32 \pm 0.44$ | $54.69 \pm 0.59$ | $72.71 \pm 0.43$ |
| PN+rot (Gidaris et al., 2019) | Conv4-512 | $56.02 \pm 0.46$ | $74.00 \pm 0.35$ | – | – |
| CC+rot (Gidaris et al., 2019) | Conv4-512 | $56.27 \pm 0.43$ | $74.30 \pm 0.33$ | – | – |
| IEPT (ours) | Conv4-512 | $\mathbf{58.43 \pm 0.46}$ | $\mathbf{75.07 \pm 0.33}$ | $\mathbf{60.91 \pm 0.59}$ | $\mathbf{79.61 \pm 0.45}$ |
| ProtoNet† (Snell et al., 2017) | ResNet-12 | $62.39 \pm 0.51$ | $80.53 \pm 0.42$ | $68.23 \pm 0.50$ | $84.03 \pm 0.41$ |
| TADAM (Oreshkin et al., 2018) | ResNet-12 | $58.50 \pm 0.30$ | $76.70 \pm 0.38$ | – | – |
| MetaOptNet (Lee et al., 2019) | ResNet-12 | $62.64 \pm 0.61$ | $78.63 \pm 0.46$ | $65.99 \pm 0.72$ | $81.56 \pm 0.63$ |
| MTL (Sun et al., 2019) | ResNet-12 | $61.20 \pm 1.80$ | $75.50 \pm 0.80$ | $65.62 \pm 1.80$ | $80.61 \pm 0.90$ |
| CAN (Hou et al., 2019) | ResNet-12 | $63.85 \pm 0.48$ | $79.44 \pm 0.34$ | $69.89 \pm 0.51$ | $84.23 \pm 0.37$ |
| AM3 (Xing et al., 2019) | ResNet-12 | $65.21 \pm 0.49$ | $75.20 \pm 0.36$ | $67.23 \pm 0.34$ | $78.95 \pm 0.22$ |
| Shot-Free (Ravichandran et al., 2019) | ResNet-12 | $59.04 \pm 0.43$ | $77.64 \pm 0.39$ | $66.87 \pm 0.43$ | $82.64 \pm 0.43$ |
| Neg-Cosine (Liu et al., 2020) | ResNet-12 | $63.85 \pm 0.81$ | $81.57 \pm 0.56$ | – | – |
| Distill (Tian et al., 2020) | ResNet-12 | $64.82 \pm 0.60$ | $82.14 \pm 0.43$ | $71.52 \pm 0.69$ | $86.03 \pm 0.49$ |
| DSN-MR (Simon et al., 2020) | ResNet-12 | $64.60 \pm 0.72$ | $79.51 \pm 0.50$ | $67.39 \pm 0.82$ | $82.85 \pm 0.56$ |
| DeepEMD (Zhang et al., 2020) | ResNet-12 | $65.91 \pm 0.82$ | $82.41 \pm 0.56$ | $71.16 \pm 0.87$ | $86.03 \pm 0.58$ |
| FEAT (Ye et al., 2020) | ResNet-12 | $66.78 \pm 0.20$ | $82.05 \pm 0.14$ | $70.80 \pm 0.23$ | $84.79 \pm 0.16$ |
| ProtoNet+Rotation (Su et al., 2020) | ResNet-18 | – | $76.00 \pm 0.60$ | – | $78.90 \pm 0.70$ |
| IEPT (ours) | ResNet-12 | $\mathbf{67.05 \pm 0.44}$ | $\mathbf{82.90 \pm 0.30}$ | $\mathbf{72.24 \pm 0.50}$ | $\mathbf{86.73 \pm 0.34}$ |

**Feature Extractors.** For fair comparison with published results, our IEPT adopts three widely-used feature extractors: Conv4-64 (Vinyals et al., 2016), Conv4-512, and ResNet-12 (He et al., 2016a). Particularly, Conv4-512 is almost the same as Conv4-64 except having a different channel size of the last convolution layer. To speed up the training process, as in many previous works (Ye et al., 2020; Zhang et al., 2020; Simon et al., 2020), we pretrain all the feature extractors on the training split of each dataset for our IEPT. Following (He et al., 2016a), we use the temperature scaling skill during the training phase. On both datasets, the input image size is $84 \times 84$. The output feature dimensions of Conv4-64, Conv4-512, and ResNet-12 are 64, 512, and 640, respectively.

**Evaluation Metrics.** We take the 5-way 5-shot (or 1-shot) FSL evaluation setting, as in previous works. We randomly sample 2,000 episodes from the test split and report the mean classification accuracy (top-1, %) as well as the 95% confidence interval. Since the integration transformer copes with each sample independently, we take a strict non-transductive setting during evaluation.

**Implementation Details.** PyTorch is used for our implementation. We utilize the Adam optimizer (Kingma & Ba, 2015) for Conv4-64 & Conv4-512 and the SGD optimizer for ResNet-12 to train our IEPT model. The hyperparameters of our IEPT model are selected according to the performance on the validation split.We will release the code soon.

## 4.2  MAIN RESULTS

**Comparison to State-of-the-Arts.** We compare our IEPT with two groups of baselines: (1) Recent SSL-based FSL methods (Gidaris et al., 2019; Su et al., 2020); (2) Representative/latest FSL methods (w/o SSL) (Snell et al., 2017; Finn et al., 2017; Lee et al., 2019; Ravichandran et al., 2019; Simon et al., 2020; Zhang et al., 2020; Ye et al., 2020; Liu et al., 2020). The comparative results for 5-way

Table 2: Ablation study results for our full IEPT model over *mini*ImageNet and *tiered*ImageNet. Our full model includes two self-supervised losses (i.e. $\mathcal{L}_{epis}$ and $\mathcal{L}_{inst}$) and two supervised losses (i.e. $\mathcal{L}_{aux}$ and $L_{integ}$). Conv4-64 is used as the feature extractor.

| $\mathcal{L}_{integ}$ | $\mathcal{L}_{inst}$ | $\mathcal{L}_{epis}$ | $\mathcal{L}_{aux}$ | *mini*ImageNet | | *tiered*ImageNet | |
|---|---|---|---|---|---|---|---|
| | | | | **1-shot** | **5-shot** | **1-shot** | **5-shot** |
| ✓ | | | | $55.04 \pm 0.52$ | $72.01 \pm 0.41$ | $56.98 \pm 0.47$ | $74.15 \pm 0.51$ |
| ✓ | ✓ | | | $55.49 \pm 0.56$ | $72.54 \pm 0.46$ | $57.41 \pm 0.51$ | $74.65 \pm 0.50$ |
| ✓ | | ✓ | | $55.88 \pm 0.43$ | $72.97 \pm 0.40$ | $57.76 \pm 0.45$ | $75.06 \pm 0.40$ |
| ✓ | ✓ | ✓ | | $55.97 \pm 0.57$ | $73.28 \pm 0.39$ | $57.83 \pm 0.55$ | $75.22 \pm 0.48$ |
| ✓ | ✓ | ✓ | ✓ | $\mathbf{56.26 \pm 0.45}$ | $\mathbf{73.91 \pm 0.34}$ | $\mathbf{58.25 \pm 0.48}$ | $\mathbf{75.63 \pm 0.46}$ |

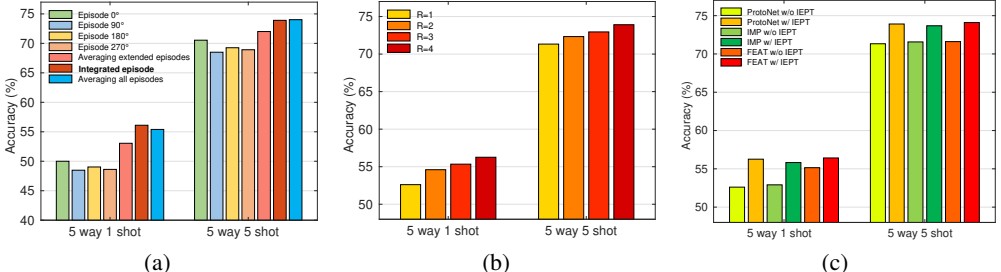

(a)                                     (b)                                     (c)

Figure 2: (**a**) Comparison among different combination methods over episodes for FSL with self-supervision. (**b**) Illustration of the effect of different choices of $R$ on the performance of our model ($R$ denotes the number of extended episodes used for SSL). (**c**) Comparative results obtained by our IEPT using different basic FSL classifiers (i.e. ProtoNet, FEAT, and IMP). It can be seen clearly that integrated episode-based fusion leads to more separation between classes. All figures present 5-way 1-shot/5-shot results on *mini*ImageNet, using Conv4-64 as the feature extractor.

1/5-shot FSL are shown in Table 1. We have the following observations: (1) When compared with the representative/latest FSL methods (w/o SSL), our IEPT achieves the best performance on all datasets and under all settings, validating the effectiveness of SSL with IEPT for FSL. (2) Our IEPT also clearly outperforms the two SSL-based FSL methods (Gidaris et al., 2019; Su et al., 2020) which only use instance-level pretext tasks, demonstrating the importance of closer/episode-level integration of SSL into FSL. (3) The improvements achieved by our IEPT over ProtoNet range from 2% to 5%. Since our IEPT takes ProtoNet as the baseline, the obtained margins provide direct evidence that SSL brings significant benefits to FSL. Note that our IEPT is also shown to be effective under both the fine-grained FSL and cross-domain FSL settings in Sec. 4.3 (see Table 3).

**Ablation Study.** Our full IEPT model is trained with four losses (see Eq. (12)), including two self-supervised losses and two supervised losses: the episode-level SSL loss $\mathcal{L}_{epis}$, the instance-level SSL loss $\mathcal{L}_{inst}$, the auxiliary FSL loss $\mathcal{L}_{aux}$ and the integrated FSL loss $L_{integ}$. To demonstrate the contribution of each loss, we present the ablation study results for our full IEPT model in Table 2, where Conv4-64 is used as the backbone. We start with $L_{integ}$ and then add the additional three losses successively. It can be observed that the performance of our model continuously increases when more losses are used, indicating that each loss contributes to the final performance.

## 4.3 Further Evaluations

**Different Combination Methods over Episodes.** We have introduced a transformer-based attention module to fuse the features of each instance from all extended episodes (and an integrated episode can be obtained) for the supervised classification task (see Sec. 3.3). In this experiment, we compare it with two alternative ways of across-episode integration: (1) Averaging extended episodes: the extended episodes are directly fused for FSL classification; (2) Averaging all episodes: the extended episodes as well as the integrated episode are fused for FSL classification. We present the comparative results on *mini*ImageNet in Figure 2(a). For comprehensive comparison, the results of FSL with each single extended episode are also reported. We can observe that: (1) The performance of 'Episode $0°$' is the highest among the four baselines (i.e., FSL with single extended episode), perhaps because the feature extractor is pretrained on the original images without rotation transformations. (2) FSL by averaging extended episodes (i.e., 'Averaging extended episodes') indeed improves each of the four

Table 3: Comparative results for the fine-grained FSL on CUB (Wah et al., 2011) and the cross-domain FSL on *mini*ImageNet → CUB.

| Method | Backbone | CUB | | *mini*ImageNet→CUB | |
| --- | --- | --- | --- | --- | --- |
| | | 1-shot | 5-shot | 1-shot | 5-shot |
| MatchingNet (Vinyals et al., 2016) | Conv4-64 | $61.16 \pm 0.89$ | $72.86 \pm 0.70$ | $42.62 \pm 0.55$ | $56.53 \pm 0.44$ |
| ProtoNet (Snell et al., 2017) | Conv4-64 | $63.72 \pm 0.22$ | $81.50 \pm 0.15$ | $50.51 \pm 0.56$ | $69.28 \pm 0.40$ |
| MAML (Finn et al., 2017) | Conv4-64 | $55.92 \pm 0.95$ | $72.09 \pm 0.76$ | $43.59 \pm 0.54$ | $54.18 \pm 0.41$ |
| Relation Net (Sung et al., 2018) | Conv4-64 | $62.45 \pm 0.98$ | $76.11 \pm 0.69$ | $49.84 \pm 0.54$ | $68.98 \pm 0.42$ |
| FEAT (Ye et al., 2020) | Conv4-64 | $68.87 \pm 0.22$ | $82.90 \pm 0.15$ | $51.52 \pm 0.54$ | $70.16 \pm 0.40$ |
| IEPT (ours) | Conv4-64 | $\mathbf{69.97 \pm 0.49}$ | $\mathbf{84.33 \pm 0.33}$ | $\mathbf{52.68 \pm 0.56}$ | $\mathbf{72.98 \pm 0.40}$ |

baselines. (3) FSL with integrated episode (i.e., 'Integrated episode') is superior to FSL by simply averaging extended episodes. (4) Comparing 'Integrated episode' with 'Averaging all episodes', the performance of FSL with integrated episode is more stable across different settings, furthering validating the usefulness of our across-episode integration. Overall, the episode-integration module is indeed effective in FSL with self-supervision. This is also supported by the visualization results presented in Appendices A.3 & A.4.

**Different Number of Extended Episodes.** In all the above experiments, the number of the extended episodes $R$ is set to 4 (rotation by $0°, 90°, 180°, 270°$). Figure 2(b) shows the impact of the value of $R$. Note that when $R = 1$, our IEPT model is equivalent to ProtoNet which is without self-supervision. It can be seen that the performance of our model consistently grows when $R$ increases from 1 to 4. Additionally, the study on exploiting other pretext tasks for our IEPT is presented in Appendix A.1.

**Different Basic FSL Classifiers.** As mentioned in Sec. 3.1, we adopt ProtoNet as the basic FSL classifier due to its scalability and simplicity. To further show the effectiveness of our IEPT when other basic FSL classifiers are used, we provide the results obtained by our IEPT using ProtoNet, FEAT, and IMP for FSL in Figure 2(c). It can be clearly observed that our IEPT leads to an improvement of about 1-4% over each basic FSL method (ProtoNet, FEAT, or IMP), indicating that our IEPT can be applied to improve a variety of popular FSL methods.

**Comparative Results for Fine-Grained FSL and Cross-Domain FSL.** To evaluate our IEPT algorithm under the fine-grained FSL and cross-domain FSL settings, we conduct experiments on CUB (Wah et al., 2011) and *mini*ImageNet → CUB, respectively. For fine-grained FSL on CUB, following (Ye et al., 2020), we randomly split the dataset into 100 training classes, 50 validation classes, and 50 test classes. For cross-domain FSL on *mini*ImageNet → CUB, the 100 training classes are from *mini*ImageNet; the 50 validation classes and 50 test classes (using the aforementioned split for fine-grained FSL) are from CUB. Under both settings, we use Conv4-64 as the feature extractor. The 5-way 1/5-shot FSL results are shown in Table 3. Our IEPT clearly achieves the best results, yielding 1–3% improvements over the second-best FEAT. This shows the effectiveness of our IEPT under both fine-grained and cross-domain settings.

## 5 CONCLUSION

We have proposed a novel Instance-level and Episode-level Pretext Task (IEPT) framework for integrating SSL into FSL. For the first time, we have introduced an episode-level pretext task for FSL with self-supervision, in addition to the conventional instance-level pretext task. Moreover, we have also developed an episode extension-integration framework by introducing an integration transformer module to fully exploit the extended episodes for FSL. Extensive experiments on two benchmarks demonstrate that the proposed model (i.e., FSL with IEPT) achieves the new state-of-the-art. Our ongoing research directions include: exploring other episode-level pretext tasks for FSL with self-supervision, and applying FSL with self-supervision to other vision problems.

ACKNOWLEDGMENTS

This work was supported in part by National Natural Science Foundation of China (61976220 and 61832017), Beijing Outstanding Young Scientist Program (BJJWZYJH012019100020098), Open Project Program Foundation of Key Laboratory of Opto-Electronics Information Processing, Chinese Academy of Sciences (OEIP-O-202006), and Alibaba Innovative Research (AIR) Program.

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

# A   APPENDIX

## A.1   COMPARISON AMONG DIFFERENT SSL STRATEGIES

To generate the extended episodes in IEPT, we apply four rotation transformations (i.e. rotation by $0°$, $90°$, $180°$, $270°$) to each image. It makes sense to explore whether other self-supervised strategies are also effective for our IEPT. To this end, we exploit shuffling image patches (see Figure 3) for self-supervised learning (SSL). Specifically, we divide each image into 2*2 patches and reorganize the patch orders to obtain a shuffling label. To compare with the rotation strategy fairly, we choose only four shuffling orders: (1, 2, 3, 4), (2, 3, 4, 1), (3, 4, 1, 2) and (4, 1, 2, 3). Note that the (1, 2, 3, 4) shuffling order equals to the original image. Similar to the rotation strategy, a fully-connected layer is utilized to recognize the shuffling order. The comparative results are shown in Table 4. We can see that both IEPT with shuffling and IEPT with rotation achieve better performance than the original ProtoNet. Particularly, IEPT with shuffling yields 1-3% and 3-4% improvements under 5-shot and 1-shot, respectively. This clearly shows the effectiveness of our IEPT for FSL even when different SSL strategies are used to define the pretext tasks.

Table 4: FSL results obtained by our IEPT using two SSL strategies (i.e. rotation and shuffling image patches). Conv4-64 is used as the feature extractor.

| Method | Backbone | *mini*ImageNet | | *tiered*ImageNet | |
|---|---|---|---|---|---|
| | | 1-shot | 5-shot | 1-shot | 5-shot |
| ProtoNet | Conv4-64 | $52.61 \pm 0.52$ | $71.33 \pm 0.41$ | $53.33 \pm 0.50$ | $72.10 \pm 0.41$ |
| IEPT (rotation) | Conv4-64 | $56.26 \pm 0.45$ | $73.91 \pm 0.34$ | $58.25 \pm 0.48$ | $75.63 \pm 0.46$ |
| IEPT (shuffling) | Conv4-64 | $55.57 \pm 0.60$ | $72.84 \pm 0.54$ | $57.81 \pm 0.48$ | $74.92 \pm 0.50$ |

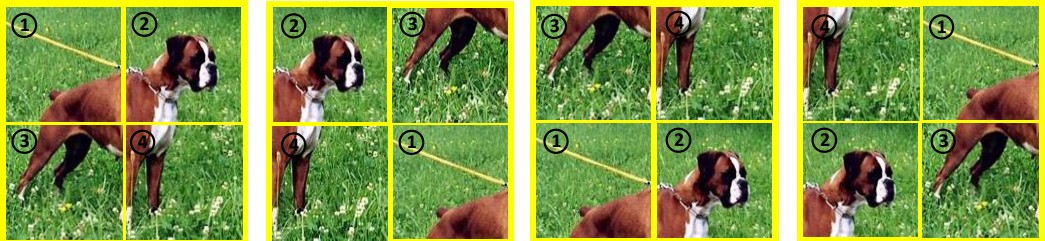

Figure 3: Illustration of the self-supervised strategy by shuffling image patches.

## A.2   COMPARISON AMONG DIFFERENT INTEGRATION APPROACHES

We employ the integration transformer to find the intrinsic correlation of various rotation-transformed instances. The transformer architecture is based on a self-attention mechanism. Concretely, it receives the feature sets of extended episodes as input $Q$, $K$ and $V$. Further, it matches each query in $Q$ with a list of keys in $K$ and returns the weighted sum of corresponding values. To show the importance of the transformer module, we compare it with two other integration approaches (i.e. concatenating and averaging) to integrate the features of extended episodes. The comparative results in Table 5 demonstrate that the integration transformer consistently performs better than the simply concatenating/averaging approaches. This suggests that the attention-based integration transformer is a better choice for designing the integration module.

Table 5: Comparative results obtained by three different approaches to integrating the features from the extended episodes, with Conv4-64 being the feature extractor.

| Method | Backbone | *mini*ImageNet | | *tiered*ImageNet | |
|---|---|---|---|---|---|
| | | 1-shot | 5-shot | 1-shot | 5-shot |
| concatenating | Conv4-64 | $51.52 \pm 0.60$ | $73.36 \pm 0.49$ | $50.78 \pm 0.68$ | $72.79 \pm 0.57$ |
| averaging | Conv4-64 | $51.58 \pm 0.62$ | $70.97 \pm 0.54$ | $53.91 \pm 0.69$ | $72.52 \pm 0.59$ |
| transformer (ours) | Conv4-64 | $\mathbf{56.26 \pm 0.45}$ | $\mathbf{73.91 \pm 0.34}$ | $\mathbf{58.25 \pm 0.48}$ | $\mathbf{75.63 \pm 0.46}$ |

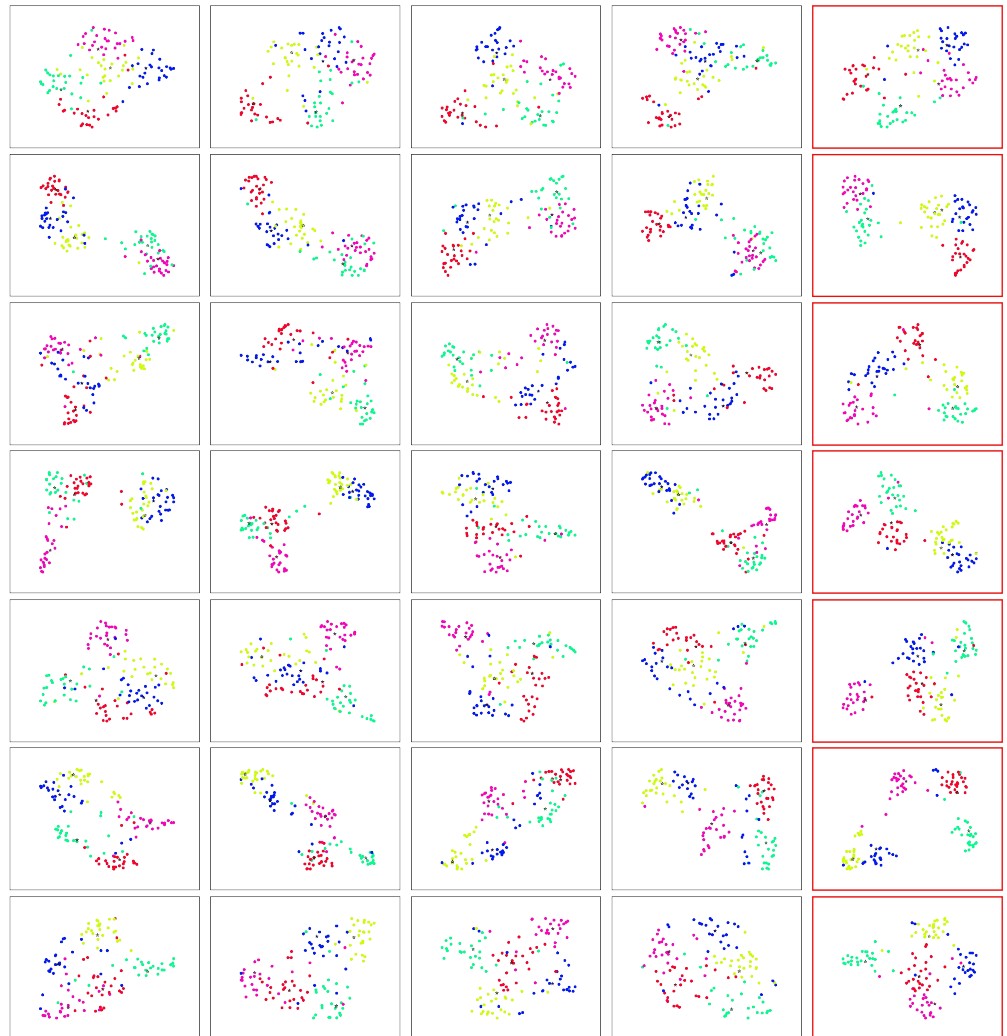

Figure 4: Feature visualizations of a number of test episodes using the UMAP algorithm (McInnes et al., 2018). Each row indicates a group of test extended episodes (the first four columns, rotation by $0°$, $90°$, $180°$, $270°$) and their integrated episode (the last column). The 5-way 5-shot FSL (with Conv4-64) is adopted on *mini*ImageNet.

### A.3 Feature Visualizations of Test Episodes

We provide the feature visualizations of test episodes in Figure 4. It can be seen that an integrated episode (the last one in each row) clearly has a better cluster data structure than the corresponding four extended episodes (the first four ones in each row). This indicates that our transformer-based across-episode integration is indeed effective for few-shot classification with self-supervision.

### A.4 Attention visualization of Test Episodes

We present attention map visualization of two test episodes (left and right) in Figure 5. Each average attention map is computed by averaging the attention map of all instances of a certain class. We can observe that: (1) The average attention maps from different classes vary significantly, showing that the diverse semantics of different classes can be reflected by our attention-based integration transformer. (2) When the classes of two episodes overlap (e.g., 'trifle' and 'dalmatian'), the average attention maps of an overlapped class from two episodes are similar, illustrating that our attention-based integration transformer can well capture the semantics of classes across episodes.

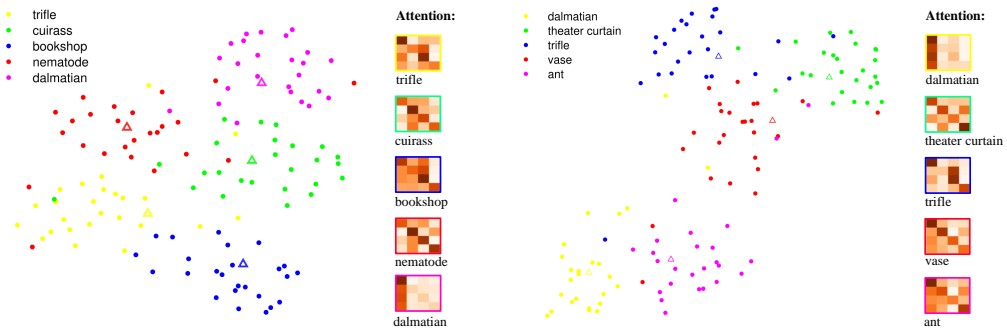

Figure 5: Feature and attention visualization of two test episodes (left and right). Both figures present 5-way 5-shot results on *mini*ImageNet, using Conv4-64 as the feature extractor.

## A.5 COMPARISON WITH SIMPLE BASELINE FOR SSL+FSL

We compare our IEPT with a simple baseline that trains the model with $\mathcal{L}_{aux} + \mathcal{L}_{inst}$ and then makes inference by just averaging the outputs of different extended episodes. The results are shown in Table 6. It can be observed that the performance of our IEPT is much more effective than that of simple integration, due to the extra use of $\mathcal{L}_{integ} + \mathcal{L}_{epis}$ for FSL.

Table 6: Comparison with the simple baseline that trains the model with $\mathcal{L}_{aux} + \mathcal{L}_{inst}$ and then makes inference by just averaging the outputs of different extended episodes.

| Method | Backbone | *mini*ImageNet | | *tiered*ImageNet | |
|---|---|---|---|---|---|
| | | 1-shot | 5-shot | 1-shot | 5-shot |
| $\mathcal{L}_{aux} + \mathcal{L}_{inst}$ | Conv4-64 | $53.25 \pm 0.46$ | $71.50 \pm 0.42$ | $55.06 \pm 0.44$ | $72.87 \pm 0.42$ |
| IEPT (ours) | Conv4-64 | $\mathbf{56.26 \pm 0.45}$ | $\mathbf{73.91 \pm 0.34}$ | $\mathbf{58.25 \pm 0.48}$ | $\mathbf{75.63 \pm 0.46}$ |

## A.6 DIFFERENT ALTERNATIVES OF SELF-SUPERVISED LOSSES

In Table 7, we provide further ablation study regarding different alternatives of $\mathcal{L}_{epis}$ and $\mathcal{L}_{inst}$. For the episode-level self-supervised loss $\mathcal{L}_{epis}$, we compare our implementation (using the KL loss between each distribution and the mean distribution) with that using a pairwise KL loss (i.e., the KL loss between each pair of distributions). For the instance-level self-supervised loss $\mathcal{L}_{inst}$, we compare our implementation (using the rotation prediction loss) with the recent self-supervised learning technique (Chen et al., 2020). We observe that our implementation achieves slight performance improvements over those using the pairwise KL loss or the contrastive learning loss.

Table 7: Ablation study results regarding different alternatives of $\mathcal{L}_{epis}$ and $\mathcal{L}_{inst}$.

| Method | Backbone | *mini*ImageNet | | *tiered*ImageNet | |
|---|---|---|---|---|---|
| | | 1-shot | 5-shot | 1-shot | 5-shot |
| IEPT ($\mathcal{L}_{inst}$-SimCLR) | Conv4-64 | $56.04 \pm 0.44$ | $73.67 \pm 0.41$ | $58.24 \pm 0.43$ | $75.59 \pm 0.41$ |
| IEPT ($\mathcal{L}_{epis}$-Pairwise) | Conv4-64 | $55.95 \pm 0.47$ | $73.72 \pm 0.40$ | $57.91 \pm 0.45$ | $75.28 \pm 0.40$ |
| IEPT (ours) | Conv4-64 | $\mathbf{56.26 \pm 0.45}$ | $\mathbf{73.91 \pm 0.34}$ | $\mathbf{58.25 \pm 0.48}$ | $\mathbf{75.63 \pm 0.46}$ |

## A.7 APPLICATION OF IEPT TO OPTIMIZATION-BASED METHOD MAML

In Table 8, we show the results obtained by applying our IEPT to the optimization-based model MAML (Finn et al., 2017). We use Conv4-64 as the feature extractor. We can see that our IEPT brings 0.7%-2.1% improvements to MAML. This further shows the flexibility (as well as effectiveness) of our IEPT for FSL.

Table 8: Comparative results by applying IEPT to the optimization-based method MAML.

| Method | Backbone | *mini*ImageNet | | *tiered*ImageNet | |
| --- | --- | --- | --- | --- | --- |
| | | 1-shot | 5-shot | 1-shot | 5-shot |
| MAML | Conv4-64 | $48.70 \pm 1.84$ | $63.10 \pm 0.92$ | $51.67 \pm 1.81$ | $70.30 \pm 0.80$ |
| MAML+IEPT | Conv4-64 | $\mathbf{49.68 \pm 0.50}$ | $\mathbf{65.22 \pm 0.48}$ | $\mathbf{52.85 \pm 0.52}$ | $\mathbf{71.04 \pm 0.49}$ |

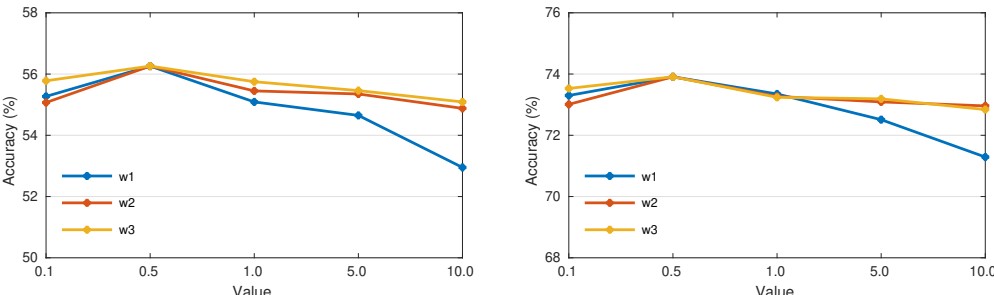

Figure 6: Visualization of our hyper-parameter analysis under 5-way 1-shot (left) and 5-shot (right) on *mini*ImageNet. Conv4-64 is used as the feature extractor.

## A.8 HYPER-PARAMETER SENSITIVITY TEST

We select the hyper-parameters $w_1$, $w_2$ and $w_3$ from the candidate set $\{0.1, 0.5, 1.0, 5.0, 10.0\}$ and show the hyper-parameter analysis results in Figure 6. We find that the performance of our IEPT is relatively stable. Concretely, the performance of our IEPT is not sensitive to $w_1$ and $w_2$ with proper values, but too large $w_1$(i.e. $w_1 = 10.0$) tends to cause obvious degradation, perhaps because the FSL task is biased by the rotation prediction loss.

