# OpenReview forum: "IEPT: Instance-Level and Episode-Level Pretext Tasks for Few-Shot Learning"
_ICLR.cc/2021/Conference — ICLR 2021 Poster_

### Official Review · AnonReviewer2 · 2020-10-28
**Official Blind Review #2**

**Rating:** 5
**Confidence:** 4

**Review:**

This paper addresses the problem of few-shot classification by incorporating self-supervised learning into the standard episode-based meta learning. Specifically, it adopts the pretext task of rotation prediction into the episode design. For each sampled episode, additional episodes are constructed by using rotated examples in the original support set and query set. Two self-supervised losses are designed based on the augmented episode sets – (1) recognizing different rotation transformations as an instance-level pretext task; (2) ensuring consistent predictions of class labels across different episodes as an episode-level pretext task. Two few-shot losses are also designed – (1) predicting class labels for each individual episode; (2) predicting class labels for the fused episode set based on attention. Experimental evaluation is conducted on two standard few-shot classification benchmarks, namely miniImageNet and tieredImageNet, and shows improved performance.

Strengths:

++ Self-supervised learning and few-shot learning are two important techniques to transfer knowledge for addressing new tasks, but their combination is less explored.

++ The proposed approach that integrates self-supervised learning into the episode design is interesting.

Suggestions and questions:

-- Compared with the state-of-the-art methods, the performance improvements of the proposed approach are marginal. For example, while the proposed approach is more complicated, it is comparable to much simpler approach like Tian et al.

-- The proposed way of integrating self-supervised learning into episode learning is claimed to be general. In addition to the metric-learning based approach ProtoNet, it would be interesting to show if the proposed approach can be applied to other types of meta-learning techniques, such as optimization based approach MAML.

-- Following the previous comment, it would be interesting to show if the proposed approach can be applied to other self-supervised learning techniques, such as the recent MoCo or simCLR models.

-- In the design of a set of extended episodes, examples within the same episode belong to the same rotation transformation. How if this is not guaranteed? That is, the examples within the episode are randomly sampled from the rotation transformations.

-- In Eq 6, how if using a pairwise KL loss without computing the mean distribution in Eq 5?

-- Why further introducing an auxiliary loss is helpful, given that the attention mechanism already fuses the information from all the episodes?

-- How is the hyperparameter sensitivity regarding different ws in Eq 12?

-- Table 2 did not provide extensive ablations regarding different combinations of the loss functions.

Post-comments to the author's response:

After reading the other reviewers’ comments and the authors’ rebuttal, I am still concerned with the novelty of the approach, experimental evaluation, and performance improvements over previous work. These issues are pointed out by the other reviewers as well. Hence, I will go with my original decision of rejecting the paper.

---

> ### Author Response · Authors · 2020-11-19
> **Response to AnonReviewer2 – Part 1/2**
>
> We’d like to thank the reviewer for the constructive comments and suggestions. Our responses are detailed below.
>
> **Q1: Compared with the state-of-the-art methods, the performance improvements of the proposed approach are marginal. For example, while the proposed approach is more complicated, it is comparable to much simpler approach like Tian et al.** \
> A1: Compared to (Tian at al., 2020) that exploits self-distillation for FSL, our IEPT achieves 0.7%-2.2% improvements on the two benchmarks (with the ResNet-12 backbone), which can generally be considered to be statistically significant according to the 95% confidence interval. Note that since the FSL performance on these two benchmarks is saturating, it is now hard for the latest/state-of-the-art competitors to consistently perform the best under all FSL settings. In contrast, IEPT beats all these competitors on the two benchmarks with three backbones (see Table 1).
>
> **Q2: It would be interesting to show if the proposed approach can be applied to other types of meta-learning techniques, such as optimization based approach MAML.** \
> A2: Good suggestion! We have added the comparison between MAML and MAML+IEPT in Appendix A.7. We also provide the comparative results in the table below. It can be observed that our IEPT is effective for the optimization-based method MAML. Note that our IEPT is also shown to bring benefits to other FSL methods (see Figure 2(c)). These observations demonstrate the effectiveness and flexibility of our IEPT.
>
> |Method| Dataset &nbsp;| &nbsp;Backbone&nbsp;|&nbsp;5-way 1-shot&nbsp;| &nbsp;5-way 5-shot|
> |--|--|:-:|:-:|:-:|
> | MAML | miniImageNet |Conv4-64|48.70$\pm$1.84|63.10$\pm$ 0.92 |
> | MAML+IEPT | miniImageNet |Conv4-64 |49.68$\pm$0.50|65.22$\pm$0.48|
> | MAML | tieredImageNet|Conv4-64|51.67$\pm$1.81|70.30$\pm$0.80|
> | MAML+IEPT | tieredImageNet|Conv4-64|52.85$\pm$0.52|71.04$\pm$0.49|
>
> **Q3: Following the previous comment, it would be interesting to show if the proposed approach can be applied to other self-supervised learning techniques, such as the recent MoCo or simCLR models.** \
> A3: Thanks. We choose to replace the $L_{inst}$ (i.e., the rotation prediction loss) with the recent SimCLR based loss, and the obtained results are presented below. We can observe that SimCLR does not lead to further improvements. Please also see Appendix A.6 for more details.
>
> |Method| Dataset &nbsp;| &nbsp;Backbone&nbsp;|&nbsp;5-way 1-shot&nbsp;| &nbsp;5-way 5-shot|
> |--|--|:-:|:-:|:-:|
> | IEPT ($L_{inst}$-SimCLR) | miniImageNet |Conv4-64|56.04$\pm$0.44|73.67$\pm$0.41 |
> | IEPT (ours) | miniImageNet |Conv4-64 |56.26$\pm$0.45|73.91$\pm$0.34|
> | IEPT ($L_{inst}$-SimCLR) | tieredImageNet|Conv4-64|58.24$\pm$0.43|75.59$\pm$0.41|
> | IEPT (ours) | tieredImageNet|Conv4-64|58.25$\pm$0.48|75.63$\pm$0.46|
>
> **Q4: In the design of a set of extended episodes, examples within the same episode belong to the same rotation transformation. How if this is not guaranteed? That is, the examples within the episode are randomly sampled from the rotation transformations.** \
> A4: Good question. After the ICLR submission, we have actually started to investigate on this. As expected, when the rotation transformations in the extended episodes are completely random, we find that the FSL performance suffers. However, if we integrate four transformed embeddings in different orders only for query samples (but the support samples keep unchanged), we observe that the FSL performance can be further improved. Since contrastive learning has to be used to cope with such embedding shuffling, it is out of the scope of this paper.
>
> **Q5: In Eq 6, how if using a pairwise KL loss without computing the mean distribution in Eq 5?** \
> A5: Thanks for the suggestion. We have now compared our implementation (i.e., the KL loss between each distribution and the mean distribution) with a pairwise KL loss (i.e., the KL loss between every two distributions), and the obtained results are presented below. We can observe that our implementation achieves slightly better performance. Please also see Appendix A.6 for more details.
>
> |Method| Dataset &nbsp;| &nbsp;Backbone&nbsp;|&nbsp;5-way 1-shot&nbsp;| &nbsp;5-way 5-shot|
> |--|--|:-:|:-:|:-:|
> | IEPT ($L_{epis}$-Pairwise) | miniImageNet |Conv4-64|55.95$\pm$0.47|73.72$\pm$0.40 |
> | IEPT (ours) | miniImageNet |Conv4-64 |56.26$\pm$0.45|73.91$\pm$0.34|
> | IEPT ($L_{epis}$-Pairwise) | tieredImageNet|Conv4-64|57.91$\pm$0.45|75.28$\pm$0.40|
> | IEPT (ours) | tieredImageNet|Conv4-64|58.25$\pm$0.48|75.63$\pm$0.46|

---

> > ### Author Response · Authors · 2020-11-19
> > **Response to AnonReviewer2 – Part 2/2**
> >
> > **Q6: Why further introducing an auxiliary loss is helpful, given that the attention mechanism already fuses the information from all the episodes?** \
> > A6: The attention mechanism focuses on learning shared information across the four extended episodes, while introducing an auxiliary loss L_{aux} for each extended episode can help to strengthen its episode-specific representation. These two learning objectives are thus complementary to each other, which is supported by the ablation study results in Table 2.
> >
> > **Q7: How is the hyperparameter sensitivity regarding different ws in Eq 12?** \
> > A7: Thanks. We have provided a hyperparameter sensitivity test in Appendix A.8. Concretely, as shown in Figure 7, the performance of our IEPT model is not much sensitivity to the value changes of the hyperparameters.
> >
> > **Q8: Table 2 did not provide extensive ablations regarding different combinations of the loss functions.** \
> > A8: Thanks. We have provided more ablations regarding different combinations of the losses in the following two tables. They have been added in Table 2 and Appendix A.5, respectively.
> >
> > |Method| Dataset &nbsp;| &nbsp;Backbone&nbsp;|&nbsp;5-way 1-shot&nbsp;| &nbsp;5-way 5-shot|
> > |--|--|:-:|:-:|:-:|
> > |$L_{integ}+L_{epis}$| miniImageNet |Conv4-64|55.88$\pm$0.43|72.97$\pm$0.40 |
> > |IEPT (ours) | miniImageNet |Conv4-64 |56.26$\pm$0.45|73.91$\pm$0.34|
> > |$L_{integ}+L_{epis}$| tieredImageNet|Conv4-64|57.76$\pm$0.45|75.06$\pm$0.40|
> > |IEPT (ours) | tieredImageNet|Conv4-64|58.25$\pm$0.48|75.63$\pm$0.46|
> >
> > |Method| Dataset &nbsp;| &nbsp;Backbone&nbsp;|&nbsp;5-way 1-shot&nbsp;| &nbsp;5-way 5-shot|
> > |--|--|:-:|:-:|:-:|
> > |$L_{aux}+L_{inst}$| miniImageNet |Conv4-64|53.25$\pm$0.46|71.50$\pm$0.42 |
> > |IEPT (ours) | miniImageNet |Conv4-64 |56.26$\pm$0.45|73.91$\pm$0.34|
> > |$L_{aux}+L_{inst}$| tieredImageNet|Conv4-64|55.06$\pm$0.44|72.87$\pm$0.42|
> > |IEPT (ours) | tieredImageNet|Conv4-64|58.25$\pm$0.48 |75.63$\pm$0.46|

---

### Official Review · AnonReviewer4 · 2020-10-30
**IEPT cohesively combines elements of Semi-Supervised Learning and Few-Shot Learning to consistently produce state-of-the-art results on Few Shot Learning tasks.**

**Rating:** 8
**Confidence:** 4

**Review:**

In Instance-Level and Episode-Level Pretext Tasks for FSL the authors present a novel method to take advantage of auxiliary prediction tasks and consistency regularization tasks which have had large success in Self-Supervised Learning settings to improve upon FSL approaches. Furthermore, the authors incorporated a transformer-based predictor to improve upon to be used with multiple augmentations of an instance to improve upon naive averaging of multiple predictions.

Empirically, the authors demonstrate the benefits of incorporating both instance and episode-level tasks by showing significant improvements over the ProtoNet approach upon which this work builds and also achieving state-of-the-art results on several tasks with different architectural backbones. Through ablations this work demonstrates the benefit of each proposed additional loss and shows robustness to choices in pretext transformation and architectural backbone.


This work is appropriately justified and explained and with satisfactory experimentation.

For section 3.3, it may help to explain that the integrated FSL task is presented as an alternative to prediction averaging. As written, the rationale for this approach is only apparent in subsequent sections.

---

> ### Author Response · Authors · 2020-11-19
> **Response to AnonReviewer4**
>
> We’d like to greatly thank the reviewer for the positive comments.
>
> **Q1: For section 3.3, it may help to explain that the integrated FSL task is presented as an alternative to prediction averaging. As written, the rationale for this approach is only apparent in subsequent sections.** \
> A1: Thanks. We have clarified this in Sec. 3.3 in the revision.

---

### Official Review · AnonReviewer1 · 2020-10-31
**The novelty is mainly on "Episode-level Pretext Task"**

**Rating:** 6
**Confidence:** 4

**Review:**

The paper proposes both Instance-level and episode-level pretext task. In comparison to existing works (Gidaris et al., 2019; Su et al., 2020), the main novelty is to design the episode-level pretext task, which enforces consistent predictions for images with different rotations.

The paper is clearly written with experiments supporting the effectiveness.
However, the novelty is limited. It is more like existing works (Gidaris et al., 2019; Su et al., 2020) plus the the regularization of consistency for images with different augmentations. However, the latter is also not new. Indeed, it has been used in [1-2], but the authors neglect them.


[1] Antti Tarvainen and Harri Valpola. Mean teachers are better role models: Weight-averaged consis- tency targets improve semi-supervised deep learning results. In NeurIPS, 2017.
[2] Samuli Laine and Timo Aila. Temporal ensembling for semi-supervised learning. arXiv, 2016.
[3] Mehdi Sajjadi, Mehran Javanmardi, and Tolga Tasdizen. Regularization with stochastic transformations and perturbations for deep semi-supervised learning. In NeurIPS, 2016.


======
Comments after rebuttal:
I know the authors develop two components for FSL, my concern is that these components are incremental and have limited novelty.
However, I admit this paper is a high quality paper in presenting its idea, organization and empirical evaluation. Hence I increase my score to accept now.

---

> ### Author Response · Authors · 2020-11-19
> **Response to AnonReviewer1**
>
> We’d like to thank the reviewer’s comments. Our responses are detailed below.
>
> **Q1: It is more like existing works (Gidaris et al., 2019; Su et al., 2020) plus the regularization of consistency for images with different augmentations. However, the latter is also not new. Indeed, it has been used in [1-3], but the authors neglect them.** \
> A1: We disagree with this comment on the novelty of this paper. Our explanations are three-fold: (1) The three works [1-3] mentioned by the reviewer focus on semi-supervised learning through utilizing consistency regularization techniques to improve the consistency across different random augmentations. However, in this work, we focus on a completely different learning problem: close integration of self-supervised learning (SSL) and few-shot learning (FSL), i.e., SSL+FSL. Although the consistency regularization itself is not new, we have designed a new self-supervised schema to seamlessly integrate it into FSL, which is not trivial given only few shots per class. (2) Compared to (Gidaris et al., 2019; Su et al., 2020) that only exploit instance-level pretext tasks for SSL+FSL, our IEPT have two new components: episode-level consistency regularization and *episode integration transformer*, so not just the former as the reviewer suggested. Please see more detailed discussion in Introduction and Related Work. Importantly, as shown in Table 1, our IEPT clearly outperforms (Gidaris et al., 2019; Su et al., 2020), validating the effectiveness of our IEPT for SSL+FSL. (3) As a flexible framework, our IEPT can even bring improvements to the latest FSL methods (e.g., FEAT), as shown in Figure 2(c). This is really impressive, since FEAT is one of the strongest competitors in FSL. Overall, we believe that our IEPT framework is of sufficient novelty in the SSL+FSL area.

---

### Official Review · AnonReviewer3 · 2020-11-02
**Solid empirical work**

**Rating:** 7
**Confidence:** 3

**Review:**

This paper presents a method for combining self-supervised learning (SSL) (in the form of predicting the rotation applied to an image) with few-shot learning (FSL) in the domain of image classification. Compared to prior work, this paper introduces -- (i) a consistency loss which ensures FSL episodes with different rotations agree in their class predictions; and (ii) an integration method which derives the label of an image from a fused representation of all its rotations. These lead to an improvement over 3 FSL benchmarks.

Strengths:
- The paper is well-written, with extensive discussion of prior and contemporary work. The technical details are presented in clear precise terms. Despite not being an expert in this area, I had no difficulty understanding the FSL setup and the new contributions of this paper.

- The experiments are also quite thorough with convincing ablation studies and several additional details in the Appendix. Overall this is solid empirical work.

- The paper presents relatively simple ideas which lead to significant improvements. Hence, it is likely to be impactful for future work looking to build on these results.

Weaknesses:
- Though effective, the novel loss terms introduced seem rather ad-hoc. There is not much understanding gained from reading the paper about why these extensions help. A large part of the improvement over the baseline ProtoNet seems to come from the integration method (based on the ablation study). This is very interesting, but could have been explored in more depth. E.g. does this approach also work when we have many training instances?

- There is not much discussion of which hyperparameters were tuned, neither how sensitive the results are to this tuning.

Other comments:
- It is not clear what the visualization in Figure 3 represents, or how it "supports the effectiveness of episode-integration module" (section 4.3). Are the figures comparable to each other?

- Why do we need the extra mean function in Eq. 6, when there is already a summation over i and r?

- Please spell out the conference names in the references.

---

> ### Author Response · Authors · 2020-11-19
> **Response to AnonReviewer3**
>
> We’d like to greatly thank the reviewer for the positive comments. Our point-to-point responses are given blow.
>
> **Q1: There is not much understanding gained from reading the paper about why these extensions help. Does this approach also work when we have many training instances?** \
> A1: We have better motivated the proposed approach in the revision. Specifically, in our IEPT framework, the episode extensions are designed with two motivations: (1) The episode-level consistency regularization module enforces the prediction consistency among the four extended episodes for FSL. This is to make the learned model more generalizable to different intra-class variations which are not captured by the few training samples in the support set. The ablation study results in Table 2 clearly demonstrate the effectiveness of episode-level consistency regularization. (2) The episode integration transformer combines the four extended episodes with self-attention for FSL. This transformer module is designed mostly for task adaptation, i.e., given a new task represented by few samples per classes and augmented by different rotated versions, it is the best way to update the feature embedding of these samples so that the classification of the query set samples can be most accurate. The ablation study results in Figure 2(a) (as well as the visualization results in Figure 3) show that the integration transformer can learn more discriminative embeddings and thus achieve significant improvements.
>
> In summary, IEPT is an example of close integration of meta-learning and self-supervised learning. It meta-learns the optimal way of exploiting data augmentation. Simply including more augmented training instances would not help. To prove this, we have now applied the same augmentation technique to ProtoNet by training it with L_aux defined over the four extended episodes and inferring by simply averaging their outputs. In this way, this ProtoNet variant has exactly the same amount of training instances as our IEPT. With Conv4-64 as the feature extractor, the accuracies for 5-way 1-shot and 5-way 5-shot on miniImageNet are 52.80% and 71.41%, respectively. In comparison to the original ProtoNet (52.61% for 5-way 1-shot and 71.33% for 5-way 5-shot), the improvements brought by having more training instances are very marginal.
>
> **Q2: There is not much discussion of which hyperparameters were tuned, neither how sensitive the results are to this tuning.** \
> A2: Thanks for the suggestion. As stated in Sec. 4.1, the hyperparameters of our IEPT model are tuned according to its validation performance. To test the hyperparameter sensitivity, we also have provided a detailed analysis in Appendix A.8. Concretely, as shown in Figure 7, the performance of our IEPT model is not much sensitive to these hyperparameters.
>
> **Q3: It is not clear what the visualization in Figure 3 represents.** \
> A3: In Figure 3, we use the UMAP algorithm to visualize the data distributions of a test episode (more test episodes are visualized in Appendix A.3). The five subfigures represent the four extended episodes and the integrated episode, respectively. Note that they are drawn with the same trained model and thus are comparable. We can clearly see that the integrated feature embeddings are stronger than those of each single extended episode in that the classes are more separable.
>
> **Q4: Why do we need the extra mean function in Eq. 6, when there is already a summation over i and r?** \
> A.4: The mean(·) in Eq. 6 calculates the KL loss between the distribution *vectors* $ p_i^r$ and $\hat{p}_i$. The summation over i and r is used to average the losses of all the query samples for all the rotation versions.
>
> **Q5: Please spell out the conference names in the references.** \
> A5: Thanks. We have done it in the revision.

---

> > ### Comment · AnonReviewer3 · 2020-11-20
> > **Response**
> >
> > Thanks for the detailed responses -- they address most of my comments, though the reasoning behind the integration transformer still seems a little ad-hoc. However, with the new results in the revision the paper seems to be even more stronger empirically, hence I will stick to my rating.

---

> > > ### Author Response · Authors · 2020-11-20
> > > **Re: Response of AnonReviewer3**
> > >
> > > Thanks for your positive comments! And thank you again for your great work on our paper.

---

### Official Review · AnonReviewer5 · 2020-11-05
**Pretext Tasks for Few-Shot Learning**

**Rating:** 5
**Confidence:** 4

**Review:**

This paper solves the problem of few-shot learning. The recent success of SSL and FSL proves that they can handle situations that few label data are provided. Motivated by this, the author proposed a novel framework IEPT that seamlessly integrates self-supervised learning methods to few-shot learning. Unlike other trivial combination of SSL and FSL methods, this paper proposed instance-level and episode-level pretext tasks to bring on closer integration. Further, this paper proposed to use transformer to integrate features from different images and augmentations. Experiments show the model achieves new SOTA.

1. In the inference phase, you use transformer to integrate embedding from both support set and query set, which seemingly makes a transductive method. Therefore, you should compare your method to other transductive FSL methods.
2. Your ablation experiments are not complete. You are supposed to give results of training with L_{integ} (Eq.(10)) and L_{epis}  (Eq.(6)).
3. There is at least a baseline method you should compare. That is you train with L_{aux} (Eq.(11)) and L_{inst} (Eq.(3)), and inference by averaging outputs (ensemble).
4. You are using multiple augmentations on each image at test time. It seems unfair to most previous tasks. It is not clear if the success is due to the ensemble effect.

---

> ### Author Response · Authors · 2020-11-19
> **Response to AnonReviewer5**
>
> We’d like to thank the reviewer for the constructive comments and suggestions. We have accordingly made changes in the revision. Our responses are detailed below.
>
> **Q1: In the inference phase, you use transformer to integrate embedding from both support set and query set, which seemingly makes a transductive method. Therefore, you should compare your method to other transductive FSL methods.** \
> A1: Sorry for the confusion. Our setting is the standard inductive setting. More specifically, as stated in Sec. 3.3, the integration transformer integrates different augmented versions of each instance *independently*. In other words, for each instance, there is no interaction with any other instances from the query set during the inference phase. This means that we take a strict *non-transductive* FSL setting. For easy understanding, we have also clarified this in Sec. 4.1.
>
> **Q2: Your ablation experiments are not complete. You are supposed to give results of training with $L_{integ}$ (Eq.(10)) and $L_{epis}$ (Eq.(6)).** \
> A2: Thanks for the suggestion. We have added the suggested ablation study in Table 2 in the revision. For clarity, we also present the ablation study results on two benchmarks in the table below. It can be seen that our IEPT model is clearly more effective than the simple integration (i.e., $L_{integ}+L_{epis}$).
>
> |Method| Dataset &nbsp;| &nbsp; Backbone &nbsp;|&nbsp; 5-way 1-shot&nbsp;| &nbsp; 5-way 5-shot|
> |--|--|:-:|:-:|:-:|
> |$L_{integ}+L_{epis}$ &nbsp;| miniImageNet |Conv4-64|55.88$\pm$0.43|72.97$\pm$0.40 |
> |IEPT (ours) | miniImageNet |Conv4-64 |56.26$\pm$0.45|73.91$\pm$0.34|
> |$L_{integ}+L_{epis}$ &nbsp;| tieredImageNet|Conv4-64|57.76$\pm$0.45|75.06$\pm$0.40|
> |IEPT (ours) | tieredImageNet|Conv4-64|58.25$\pm$0.48|75.63$\pm$0.46|
>
> **Q3: There is at least a baseline method you should compare. That is you train with $L_{aux}$ (Eq.(11)) and $L_{inst}$ (Eq.(3)), and inference by averaging outputs (ensemble).** \
> A3: Good suggestion! We have added the suggested comparison in Appendix A.5 in the revision. We also present the obtained results here in the table below. We can observe that our IEPT model achieves 2%-3% improvements over this simple baseline (i.e., $L_{aux}+L_{inst}$), indicating the importance of exploiting $L_{integ}+L_{epis}$ for FSL.
>
> |Method| Dataset &nbsp;| &nbsp;Backbone&nbsp;|&nbsp;5-way 1-shot&nbsp;| &nbsp;5-way 5-shot|
> |--|--|:-:|:-:|:-:|
> |$L_{aux}+L_{inst}$ &nbsp; | miniImageNet |Conv4-64|53.25$\pm$0.46|71.50$\pm$0.42 |
> |IEPT (ours) | miniImageNet |Conv4-64 |56.26$\pm$0.45|73.91$\pm$0.34|
> |$L_{aux}+L_{inst}$ &nbsp;| tieredImageNet|Conv4-64|55.06$\pm$0.44|72.87$\pm$0.42|
> |IEPT (ours) | tieredImageNet|Conv4-64|58.25$\pm$0.48 |75.63$\pm$0.46|
>
> **Q4: You are using multiple augmentations on each image at test time. It seems unfair to most previous tasks. It is not clear if the success is due to the ensemble effect.** \
> A4: Thanks for pointing this out. We’d like to make the following points: (1) Note that data augmentation/deformation during inference is a standard practice. In particular, many compared methods in Table 1 use inference-time data augmentation (e.g., center crop). (2) It is also true that these compared baselines do not use multiple data augmentations and fuse the results as our method does. To make the comparison absolutely fair, we have now applied the same augmentation technique to ProtoNet by training it with L_aux defined over the four extended episodes and inferring by simply averaging their outputs. With Conv4-64 as the feature extractor, the accuracies for 5-way 1-shot and 5-way 5-shot on miniImageNet are 52.80% and 71.41%, respectively. In comparison to the original ProtoNet (52.61% for 5-way 1-shot and 71.33% for 5-way 5-shot), the improvements brought by data augmentation are very marginal. This means that directly adding multiple data augmentation and fusing the results during inference help little. (3) The key is thus to meta-learn the best way to exploit the multiple augmentations so that during inference the model can benefit from them given a new task. That is exactly what our IEPT model is designed for.

---

### Decision · Program_Chairs · 2021-01-07
**Final Decision**

**Decision:**

Accept (Poster)

**Comment:**

The submission proposes instance-level and episode-level pretext tasks as an unsupervised data augmentation mechanism for few-shot learning. Furthermore, transformer are proposed to integrate features from different images and augmentations. The paper received one clear accept, one accept, one borderline accept and two borderline reject recommendations. The main concerns of the R5 and R2 were weak ablation study and the lack of a clear advantage of the method in terms of results compared to the prior state of the art. In the rebuttal, the authors provided more ablation studies. Similarly, the reviewers were concerned about the novelty of the paper being incremental compared to the prior works. Based on the majority vote, the meta reviewer recommends acceptance.